# Targeted Inhibition of O-Linked β-N-Acetylglucosamine Transferase as a Promising Therapeutic Strategy to Restore Chemosensitivity and Attenuate Aggressive Tumor Traits in Chemoresistant Urothelial Carcinoma of the Bladder

**DOI:** 10.3390/biomedicines10051162

**Published:** 2022-05-18

**Authors:** Hye Won Lee, Mi Ju Kang, Young-Ju Kwon, Sama Abdi Nansa, Eui Hyun Jung, Sung Han Kim, Sang-Jin Lee, Kyung-Chae Jeong, Youngwook Kim, Heesun Cheong, Ho Kyung Seo

**Affiliations:** 1Department of Urology, Center for Urologic Cancer, National Cancer Center, Goyang 10408, Korea; uroproper@ncc.re.kr (H.W.L.); jeh0315@ncc.re.kr (E.H.J.); 12112@ncc.re.kr (S.H.K.); 2Research Institute, National Cancer Center, Goyang 10408, Korea; 75837@ncc.re.kr (M.J.K.); mysylph@naver.com (Y.-J.K.); leesj@ncc.re.kr (S.-J.L.); jeongkc@ncc.re.kr (K.-C.J.); 3Department of Cancer Biomedical Science, Graduate School of Cancer Science and Policy, National Cancer Center, Goyang 10408, Korea; samaabdi@ncc.re.kr (S.A.N.); youngwkim@ncc.re.kr (Y.K.)

**Keywords:** urothelial carcinoma of bladder, chemoresistance, gemcitabine, paclitaxel, O-linked N-acetylglucosaminylation, O-linked β-N-acetylglucosamine transferase, biomarkers

## Abstract

Acquisition of acquired chemoresistance during treatment cycles in urothelial carcinoma of the bladder (UCB) is the major cause of death through enhancing the risk of cancer progression and metastasis. Elevated glucose flux through the abnormal upregulation of O-linked β-N-acetylglucosamine (O-GlcNAc) transferase (OGT) controls key signaling and metabolic pathways regulating diverse cancer cell phenotypes. This study showed that OGT expression levels in two human UCB cell models with acquired resistance to gemcitabine and paclitaxel were significantly upregulated compared with those in parental cells. Reducing hyper-O-GlcNAcylation by OGT knockdown (KD) markedly facilitated chemosensitivity to the corresponding chemotherapeutics in both cells, and combination treatment with OGT-KD showed more severe growth defects in chemoresistant sublines. We subsequently verified the suppressive effects of OGT-KD monotherapy on cell migration/invasion in vitro and xenograft tumor growth in vivo in chemoresistant UCB cells. Transcriptome analysis of these cells revealed 97 upregulated genes, which were enriched in multiple oncogenic pathways. Our final choice of suspected OGT glycosylation substrate was VCAN, S1PR3, PDGFRB, and PRKCG, the knockdown of which induced cell growth defects. These findings demonstrate the vital role of dysregulated OGT activity and hyper-O-GlcNAcylation in modulating treatment failure and tumor aggression in chemoresistant UCB.

## 1. Introduction

Cytotoxic chemotherapeutics play a pivotal role in treating locally advanced, metastatic, and localized urothelial carcinoma of the bladder (UCB) as multiple or single agents [1,2,3,4,5,6,7,8,9,10,11,12,13]. Gemcitabine (GEM), a derivative of cytidine, can inhibit DNA synthesis, cause DNA double-strand breaks, and arrest cells at the G1/S stage, resulting in cell death [13]. Regimens containing GEM are used intravesically to prevent nonmuscle-invasive UCB recurrence, even in Bacillus Calmette–Guérin unresponsive patients, and systemically as both first- and second-line chemotherapy for locally advanced or metastatic UCB [1,2,3,4,5,6,7,8,9,10,11,12]. Another agent, paclitaxel (PTX), interferes with microtubule dynamics and activates the mitotic spindle-assembly checkpoint to induce mitotic arrest and cell death [5]. PTX-based chemotherapy regimens are still useful in clinical practice against nonmuscle-invasive UCB and locally advanced or metastatic UCB following the failure of a platinum-based regimen [5,7]; however, intrinsic or acquired chemoresistance in UCB has long been an obstacle to achieving a cure, and treatment failure contributes to a high risk of progression to more aggressive tumors capable of metastasizing to distant sites and, ultimately, patient mortality [5,14,15,16,17,18,19,20,21,22].

As GEM- and PTX-resistant UCBs have also been associated with multidrug resistance (MDR), a phenomenon in which cancer cells develop cross-resistance to different chemotherapeutic drugs following extensive exposure [5,14,15,20,23], it is important to elucidate the precise mechanisms of action related to chemoresistance to GEM and PTX, as doing so may aid identification of alternative or sensitizing therapies for chemoresistant UCB.

Similar to most cancers, UCB cells acquire glucose and glutamine-derived carbon sources to fuel ATP synthesis through aerobic glycolysis (Warburg effect) to promote their proliferation and survival [24,25,26]. The elevated metabolic flux through the hexosamine biosynthetic pathway in cancer cells contributes to the increased production of uridine diphosphate N-acetylglucosamine (UDP-GlcNAc), which is the substrate of O-linked β-N-acetylglucosamine (O-GlcNAc) transferase (OGT) [24,25,26,27]. OGT-mediated O-GlcNAcylation is an inducible and reversible post-translational modification (PTM) that influences protein biological functions, conformation and stability, and intracellular localization, potentially by providing a recognition platform for subsequent protein–protein interactions (PPIs) [24,26,28,29,30,31,32]. Subsequent upregulation of OGT activity in cancer cells is increasingly considered a hallmark of multiple cancers and contributes to sustaining proliferative signals, enabling replicative immortality, resisting cell death, activating invasion and metastasis, inducing angiogenesis, and deregulating cellular energetics [24,25,26,27,28,30]. Notably, various groups have shown the contribution of upregulated OGT and hyper-O-GlcNAcylation to unfavorable pathology and tumor aggressiveness in UCB [24,26,27,28,33]. However, a comprehensive elucidation of the functional mechanisms through which O-GlcNAc promotes chemoresistance in UCB has not yet been conducted.

In this regard, integrative comparative biological, molecular, and genomic characterization achieved by comparing samples from chemoresistant UCB with paired pretreated tumors could elucidate the frequency of key genetic alterations that drives chemoresistance as well as potential therapeutic targets to resensitize tumors to chemotherapy. As the possibility of obtaining serial tumor biopsies from UCB patients undergoing chemotherapy is limited by technical and ethical barriers in clinical settings, drug-adapted UCB cell models have been successfully used to elucidate chemoresistance mechanisms [16,22,34]. For example, we previously addressed the functional role of c-Myc in the development of GEM resistance in UCB using GEM-resistant KU19-19 cells, with the results demonstrating that c-Myc inhibitors are a viable treatment option when MDR is involved [16]. In the present study, we applied experimental and bioinformatics tools to investigate the critical oncogenic roles of OGT and O-GlcNAc, particularly in human UCB cell lines with acquired chemoresistance to GEM and PTX, and evaluated the therapeutic value of targeting OGT to facilitate the chemosensitivity of chemoresistant UCB cells to their corresponding drugs.

## 2. Materials and Methods

### 2.1. Cell Lines

Human bladder epithelial cells (HBlEpC) (purchased from CELLnTEC Advanced Cell Systems, Bern, Switzerland) were maintained at 5% CO_2_ and 37 °C in epithelial cell growth medium (Cat# MD-0041; iXCells Biotechnologies, San Diego, CA, USA) supplemented with 10% fetal bovine serum (FBS; HyClone, Logan, UT, USA), 100 U/mL penicillin, and 100 μg/mL streptomycin (Gibco, Waltham, MA, USA). Human UCB cell lines RT4, 253J, and J82 were procured from the Korean Cell Line Bank (Seoul, Korea) and the human UCB cell line KU19-19 was donated by Dr. Ozu (Tokyo Medical University, Tokyo, Japan) [16]. The 235J, J82, and KU19-19 cells were maintained in RPMI 1640 medium (HyClone) supplemented with 10% FBS (HyClone, Logan, UT, USA), 100 U/mL penicillin, and 100 μg/mL streptomycin (Invitrogen, Carlsbad, CA, USA). The RT4 cells were maintained in Dulbecco’s modified Eagle’s medium (DMEM; HyClone) supplemented with 10% FBS (HyClone), 100 U/mL penicillin, and 100 μg/mL streptomycin (Invitrogen, Carlsbad, CA, USA).

Two human UCB cell lines, UMUC-3 and T24, were purchased from ATCC (Old Town Manassas, VA, USA) and all the cells were maintained at 5% CO_2_ and 37 °C with appropriate constant humidity. To establish representative UCB cell sublines with acquired chemoresistance, UMUC-3 and T24 cells were incubated with different concentrations of drugs (UMUC-3 for GEM and T24 for PTX) that resulted in a death rate of 20% during incubation of 72 h. The drug resistance of each cell line increased with exposure to progressively increasing concentrations of GEM (2–20 nM) for UMUC-3 and PTX (1–1000 nM) for T24, and we ultimately confirmed that a ≥10-fold drug resistance was acquired in both cell lines. UMUC3 and UMUC-3-GEMr cells were cultured in DMEM (HyClone) supplemented with 10% FBS (HyClone), 100 U/mL penicillin, and 100 μg/mL streptomycin (Gibco). T24 and T24-PTXr cells were cultured in RPMI 1640 medium (HyClone) supplemented with 10% FBS (HyClone), 100 U/mL penicillin, and 100 μg/mL streptomycin (Gibco). Authentication of these cell lines was conducted by short tandem-repeat profiling to exclude cross-contamination between cell lines. Mycoplasma contamination was examined in the cell line stocks before the study.

### 2.2. Gene Silencing with siRNA

Negative control small interfering RNA (siRNA; nontargeting pool) and siRNA targeting the genes of interest were purchased from Genolution Inc. (Seoul, Korea). The following siRNA sequences were used for the indicated target genes: siControl, 5′-CUCGUGCCGUUCCAUCAGGUAGUU-3′; si-OGT, 5′-GCAACUACUCAGAUCAACAUU-3′ (#1) and 5′- CAGAAGAUGCCAUCGUAUAUU-3′ (#2); si-versican (VCAN), 5′-GGAAGAUGGGCUAUACCUAUU-3′ (#1) and 5′-CUGCAAUACGAGAAUUGGAUU-3′ (#2); si-sphingosine-1-phosphate receptor 3 (S1PR3), 5′-CCUUACGACGCCAACAAGAUU-3′ (#1) and 5′-CUUCAGAAUGGGAUCUUCUUU-3′ (#2); si-platelet-derived growth factor receptor β (PDGFRB), 5′-CAUUCCAUGCCGAGUAACAUU-3′ (#1) and 5′-CCAUCAUCUCCCUUAUCAUUU-3′ (#2); and si-protein kinase Cγ (PRKCG), 5′-CUGACUUUGGCAUGUGUAAUU-3′ (#1) and 5′-GGAAUGAGACCUUUGUGUUUU-3′ (#2).

### 2.3. Generation of Stable Cell Lines with Viral Transduction

To generate OGT-knockdown (KD) cell lines, a lentiviral vector (pLKO.1; Addgene, Watertown, MA, USA) expressing short-hairpin (sh)RNA against *OGT* was constructed. The following shRNA sequences were used for the construct: forward, 5′-CCGGCCAAACTTTCTGGATGCTTATCTCGAGATAAGCATCCAGAAAGTTTGGTTTT-3′ and reverse, 5′-AATTCAAAAACCAAACTTTCTGGATGCTTATCTCGAGATAAGCATCCAGAAAGTTTGG-3′. Stable KD cells were generated using a lentiviral vector harboring either shOGT or scrambled shRNA as a control according to standard protocols for viral transduction.

### 2.4. Quantitative Real-Time Reverse Transcription PCR (qRT-PCR)

Total RNA was isolated using TRIzol (QIAGEN, Hilden, Germany), and cDNA was reverse transcribed from 1 µg of RNA using SuperScript IV reverse transcriptase (18090050; Invitrogen). The qPCR was performed with SYBR Green using a real-time PCR LightCycler (Roche Diagnostics, Sydney, Australia). The relative amount of cDNA was calculated by the comparative Ct method using the 18S ribosomal RNA sequence as control. The primer sequences were as follows: *OGT* (forward, CAGCATCCCAGCTCACTT and reverse, CAGCTTCACAGCTATGTCTTC); *PDGFRB* (forward, GATGCCGAGGAACTATTCATCT and reverse, TTTCTTCTCGTGCAGTGTCAC); *PRKCG* (forward, GCCACTAGGTGTCCCCAA and reverse, GAGAATATCGGGCTCCGCTC); *S1PR3* (forward, CACCCGCTAGGATGCCG and reverse, CTCCAGCGAGGGCGTTG); and *VCAN* (forward, GAGATAAGATGGGAAAGGCAGG and reverse, GGGGACAGTGAGGTGGAACA).

### 2.5. Western Blot (WB)

Cells were harvested in ice-cold radioimmunoprecipitation assay lysis buffer (50 mM Tris-HCl (pH 7.4), 150 mM NaCl, 1% NP-40, 0.5% Na-deoxycholate, 0.1% sodium dodecyl sulfate (SDS), and 1 mM EDTA) containing a protease-inhibitor cocktail (Roche Applied Bioscience, Basel, Switzerland) and a phosphatase inhibitor (Sigma-Aldrich, St. Louis, MO, USA). Soluble lysate fractions were isolated by centrifugation at 20,000× *g* for 20 min at 4 °C and quantified using a Pierce BCA Protein Assay Kit (Thermo Fisher Scientific, Waltham, MA, USA). Samples were resolved by SDS polyacrylamide gel electrophoresis using equal concentrations of protein and transferred to polyvinylidene fluoride membranes, which were blocked with 5% skim milk and then probed with the indicated primary and secondary antibodies according to standard protocols. Immunoblotted protein intensities were quantified using ImageJ software (NIH, Bethesda, MD, USA) and normalized to the loading control.

### 2.6. Antibodies and Reagents

Primary antibodies against OGT (24083) and O-GlcNAc (9875) were purchased from Cell Signaling Technology (Danvers, MA, USA). Antibodies against β-actin (A300–491A) were purchased from Bethyl Laboratories (Montgomery, TX, USA), and those against PRKCG (SC-166451) and PDGFRB (SC-374573) were purchased from Santa Cruz Biotechnology (Dallas, TX, USA). Secondary antibodies against horseradish peroxidase-linked anti-rabbit (A120–101P) and anti-mouse (A90–116P) antibodies were purchased from Bethyl Laboratories. For immunohistochemistry (IHC), primary antibodies against OGT (ab96718) and Ki67 (ab15580) were purchased from Abcam (Cambridge, UK), and antibodies against Caspase-3 (9661) were purchased from Cell Signaling Technology. GEM (G6423) was purchased from Selleck Chemicals (Houston, TX, USA). PTX (T7402), Thiamet G (SML0244), MG132 (M7449), and phosphatase inhibitor cocktails 2 and 3 were purchased from Sigma-Aldrich. MTT (M1415) was purchased from Duchefa Biochemie (Haarlem, The Netherlands). Protease-inhibitor cocktail tablets were purchased from Roche Applied Biosciences, and RNAi-Max (13778150) was purchased from Thermo Fisher Scientific.

### 2.7. Cell Growth Assay

To evaluate the cell growth rate, we performed a 3-(4,5-dimethylthiazol-2-yl)-2,5-diphenyltetrazolium bromide (MTT) assay (Duchefa Biochemie). Cells were plated in 96-well plates for 24 h and incubated with the indicated doses of either chemotherapeutic drugs or a vehicle for an additional 72 h before harvest. Subsequently, MTT reagent (0.5 mg/mL) was added to each well and incubated for 6 h. Absorbance was measured according to manufacturer instructions, and cell growth was calculated as the ratio of the absorbance after reagent treatment to absorbance after vehicle treatment.

### 2.8. Cell Proliferation Assay

Cell proliferation was measured using an image-based cell proliferation analyzer (IncuCyte^TM^; Essen Instruments, Ann Arbor, MI, USA). Cells were cultured in nutrient complete DMEM in multiwell plates overnight and imaged throughout the indicated time. An IncuCyte automated cell proliferation detector was used to measure cell proliferation through quantitative kinetic processing metrics derived from time-lapse image acquisition and presented as a percentage of cell confluence over time.

### 2.9. Wound Healing Assay

Cells were plated in 12-well plates and transfected with siRNA (50 nM) using RNAiMax reagent (Thermo Fisher Scientific, Waltham, MA, USA). After 48 h of culture, the monolayer was scraped with a sterile 200-μL pipette tip when the cells reached ~80% confluence. Subsequently, the medium was replaced, and the cells were transfected with siRNA (50 nM). Cells that migrated into the wound field were photographed using the image-based high-content screening system Operetta CLS (PerkinElmer, Waltham, MA, USA) at the time of wounding and then every 6 h for 2 days. The ratio of cell migration was calculated by measuring the length of the remaining cell-free area using Harmony High-Content Imaging and Analysis Software (PerkinElmer, Waltham, MA, USA).

### 2.10. Transwell Invasion Assay

An in vitro cell invasion assay was performed using Transwell (3422; Corning Inc., Corning, NY, USA). The upper chamber of the Transwell was coated with Matrigel (354234; Corning Inc.) diluted in a serum-free medium. Cells were resuspended in serum-free medium and seeded in the upper chamber of the Transwell, and the lower chamber was filled with fresh medium containing 10% FBS. After 40 h, the cells were removed from the upper surface of the membrane, and cells on the lower surface were fixed to the membrane using 100% methanol for 10 min, stained with 0.05% crystal violet for 30 min, and rinsed with 10% phosphate-buffered saline (PBS; HyClone). After photographing the invasive cells, the membranes were soaked in 0.05 M Na_2_HPO_4_ in 50% ethanol solution for 20 min to solubilize the crystal violet dye from the strained cells. The number of invading cells was quantified by measuring the absorbance at 590 nm using an enzyme-linked immunosorbent assay reader (Molecular Devices, Sunnyvale, CA, USA).

### 2.11. In Vivo Tumor Growth Assay

Female BALB/C nude mice (6 weeks old; Orient Bio Inc., Seongnam, Korea) were handled using aseptic procedures and allowed to adjust to local conditions for 1 week before conducting experimental manipulations. UMUC-3-GEMr cells transfected with shCTL and shOGT (5 × 10^6^/mouse) were mixed at a 1:1 dilution with Matrigel (354234; Corning Inc.) and injected subcutaneously into both flanks of each mouse at a total final volume of 100 μL (*n* = 5 mice/group). Tumor growth was evaluated by measuring the two perpendicular diameters of the tumors, and tumor size was calculated using the formula 4π/3 × (width/2)^2^ × (length/2). Tumors were harvested at the experimental endpoint. Animal experiments were conducted in accordance with the protocols approved by the Institutional Animal Care and Use Committee of the National Cancer Center, Republic of Korea (IRB No. NCC-20-585). The methods applied in this study were conducted in accordance with approved guidelines.

### 2.12. IHC Analysis

For IHC analysis of the xenograft samples from BALB/C nude mice, dissected tissues were fixed immediately after removal in 10% buffered formalin solution for a maximum of 24 h at room temperature before being dehydrated and paraffin-embedded under vacuum. Tissue sections were deparaffinized using EZ Prep buffer (Ventana Medical Systems, Santa Clara, CA, USA). Antigen retrieval was performed using CC1 buffer (Ventana Medical Systems), and sections were blocked for 30 min with background Buster solution (Innovex, Lincoln, RI, USA). IHC was performed using Discovery XT processor (Ventana Medical Systems). All tumor tissues were harvested from mice and fixed in 4% paraformaldehyde overnight. The fixed tissues were dehydrated, embedded in paraffin, and sliced into 3 μm sections, which were subsequently deparaffinized with EZ Prep buffer, and antigen retrieval was performed with CC1 buffer along with heat treatment in citrate buffer (pH 6.0; Ribo CC; Ventana Medical Systems). Tumor sections were incubated with the indicated primary antibodies, and detection was performed using a 3,3′-diaminobenzidine (DAB)-detection kit (Ventana Medical Systems) according to manufacturer instructions, followed by counterstaining with hematoxylin (Ventana Medical Systems). Images were obtained using a Vectra Polaris microscope (PerkinElmer). IHC DAB intensities were quantified using ImageJ software (NIH) and normalized to the nucleus area.

### 2.13. RNA Extraction and Whole Transcriptome Sequencing Analysis

Total RNA was isolated from UMUC3, UMUC-3-GEMr, T24, and T24-PTXr cells using the QIAGEN RNeasy Mini Kit (Invitrogen) according to manufacturer instructions, and the total RNA concentration was calculated using Quant-IT RiboGreen (#R11490; Invitrogen). To assess the integrity of the RNA, samples were run on the TapeStation RNA screentape (#5067-5575; Agilent Technologies, Santa Clara, CA, USA), and only high-quality RNA preparations with an RNA integrity number > 7.0 were used to construct the RNA library. A library was independently prepared with 1 μg of total RNA from each sample using the Illumina TruSeq Stranded mRNA Sample Prep Kit (#RS-122-2101; Illumina, San Diego, CA, USA). The first step in the workflow involved purifying the poly-A-containing mRNA molecules using poly-T-attached magnetic beads. Following purification, mRNA was fragmented into small pieces using divalent cations at elevated temperatures, and the cleaved RNA fragments were copied into first-strand cDNA using SuperScript II reverse transcriptase (#18064014; Invitrogen) and random primers. This was followed by second-strand cDNA synthesis using DNA polymerase I, RNase H, and dUTP. These cDNA fragments then went through an end-repair process, the addition of a single ‘A’ base, and ligation of the adapters. The products were then purified and enriched by PCR to create the final cDNA library. The libraries were quantified using KAPA Library Quantification kits for Illumina Sequencing platforms according to the qPCR Quantification Protocol Guide (#KK4854; Kapa Biosystems, Woburn, MA, USA) and qualified using TapeStation D1000 ScreenTape (#5067-5582; Agilent Technologies). Indexed libraries were then submitted to an Illumina NovaSeq platform (Illumina) [23], and paired-end (2 × 100 bp) sequencing was performed by Macrogen (Seoul, Korea). The reads were aligned to the hg19 genome using bowtie2 (http://bowtie-bio.sourceforge.net/index.shtml (accessed on 23 April 2022), San Diego, CA, USA) [35] or hisat2 (http://daehwankimlab.github.io/hisat2/ (accessed on 23 April 2022), Dallas, TX, USA) [36], and read abundance was estimated using either tophat2 (https://ccb.jhu.edu/software/tophat/index.shtml, accessed on 10 October 2020) [37] or stringtie (https://ccb.jhu.edu/software/stringtie/, accessed on 10 October 2020) [38], depending on whether bowtie2 or hisat2 was an aligner. The quantified raw gene count was normalized to fragments per kilobase million and log_2_ transformed for further analysis [39]. Differentially expressed genes (DEGs) between the defined groups of samples were identified using the DESeq2 R package (https://www.bioconductor.org/packages//2.10/bioc/html/DESeq.html, accessed on 15 November 2020). The Wald test for hypothesis testing was used to produce the gene list after ranking by *p* value and the *p* value was adjusted by multiple hypothesis testing using the Benjamini–Hochberg method (*Padj*, false discovery rate (FDR)) [23,40]. Transcripts with a log_2_ fold change ≥ 1 and *Padj* < 0.05 were categorized as DEGs. In addition, the Kyoto Encyclopedia of Genes and Genomes (KEGG) program was employed, and the significant pathways were determined using a multiple hypothesis Bonferroni-corrected value of *p* < 0.05 and FDR < 0.20 as cut-off criteria. The sequence data were deposited in the NCBI Sequence Read Archive (accession No. PRJNA805282).

### 2.14. Calculation of the O-GlcNAcylation Score

The O-GlcNAcylation score was calculated for each protein identified in the database of O-GlcNAcylated proteins using the O-GlcNAc Database (www.oglcnac.mcw.edu, accessed on 3 March 2021) [41] to predict the critical targets of OGT among the DEGs. The score was calculated based on the following equation, in which all components were normalized by their maximal value in the dataset of 5072 entries
S(x) = R(x)norm + C(x)norm + T(x)norm + fA(x)norm + lA(x)norm + B(x)norm
where R is the length of the list of references (Nx); C is the sum of per-year citations for each index i of Nx; T is the time between the first and last reference publications; fA and IA are the number of distinct first and last authors, respectively, within Nx; and B is a bonus term computed for each index, I, of Nx and averaged over R. Briefly, given x (the list of all protein entries) and x (a single entry), and considering the list of references, Nx, and P, the number of protein entries documented in an index, i of Nx, is determined. Furthermore, a higher Pi value negatively impacts B, whereas a higher Ci value positively impacts Bi. This absolute score is theoretically contained within a [0, 6] range, and the score is then converted on a relative scale by normalizing the score values to the top score protein entry.

### 2.15. Immunoprecipitation

We assessed whether PDGFRB and PRKCG3 were putative OGT substrates in UMUC-3-GEMr cells by confirming the existence of O-GlcNAc motifs in immunoprecipitated PDGFRB and PRKCG3 from UMUC-3-GEMr cells [42,43]. UMUC-3-GEMr cells were rinsed in ice-cold PBS and lysed in a lysis buffer containing 1% NP-40, 20 mM Tris-HCl, 150 mM NaCl, 10% glycerol, 2 mM EDTA, 10 mM NaF, 1 mM Na_3_O_4_V, 0.2 mM PMSF), a protease-inhibitor cocktail (Roche Applied Bioscience), and 1% phosphatase inhibitor cocktail (Sigma-Aldrich). Subsequently, 1 mg of the lysates for immunoprecipitation were incubated with 2 μg of the primary antibodies (anti-PRKCG antibody (Santa Cruz Biotechnology), anti-PDGFRB antibody (Santa Cruz Biotechnology), and mouse IgG (Sigma-Aldrich), respectively) at 4 °C with overnight shaking after adding 50 μL of the protein A/G agarose beads (GenDEPOT, Katy, TX, USA). The immunoprecipitates were washed three times with wash buffer and eluted by boiling in SDS sample buffer with β-mercaptoethanol for 5 min. The immunoprecipitated complex was then analyzed by immunoblotting with the indicated antibodies.

### 2.16. Statistical Analyses

Data are expressed as the mean ± standard deviation or standard error of the mean obtained from at least three independent experiments. Statistical significance was calculated using Student’s *t* test in GraphPad Prism software (v8.0; GraphPad Software, La Jolla, CA, USA), and *p* < 0.05 was considered significant.

## 3. Results

### 3.1. OGT Expression and Activity Are Elevated in Established UCB Cells with Acquired Chemoresistance to GEM and PTX

First, we validated the tumor-specific upregulation of the transcriptional and translational levels of OGT (tumor-specific upregulation of OGT/O-GlcNAc) in various UCB cell lines (Figure 1A) [26,27,33]. The representative human UCB cell lines recapitulating locally advanced and metastatic UCB (UMUC-3 and T24) [22,44] were used to investigate the contribution of OGT in promoting MDR in UCB. We then confirmed the role of OGT in UMUC-3 and T24 cells by demonstrating the effects of specific *OGT* silencing using siRNA (Figure 1B) or shRNA (Appendix A) to suppress cell growth (Figure 1C) (*p* < 0.01), migration (Figure 1D) (*p* < 0.01), and invasion (Figure 1E) (*p* < 0.001), as well as the in vivo tumorigenicity of UMUC-3 cells (Figure 1F). The results were consistent with published observations [24,26,27,28,33], which demonstrated the reliability and reproducibility of our preclinical platform.

We subsequently established two sublines with acquired resistance to GEM (UMUC-3-GEMr) or PTX (T24-PTXr) through a cell line-adaptation approach involving chronic repeated exposure to GEM and PTX through gradient culture [16,22,34]. The ratio of the 50% inhibitory concentration (IC_50_) value of each resistant line to its respective parental cell line was estimated as the degree to which GEM and PTX induced resistance. Although there were neither evident morphological changes (Appendix A) nor significant differences in cell growth kinetics (Appendix A) between UMUC-3-GEMr and T24-PTXr and their parental cell lines, UMUC-3-GEMr cells (IC_50_: 10 μM) were >500-fold less sensitive to GEM than UMUC-3 cells (IC_50_: 0.02 μM), whereas T24-PTXr cells (IC_50_: 1.37 μM) were 57-fold less sensitive to PTX than T24 cells (IC_50_: 0.024 μM) (Figure 2A). Notably, the mRNA expression levels of *OGT* (Figure 2B) and global levels of O-GlcNAc (Figure 2C) in the UMUC-3-GEMr and T24-PTXr sublines were significantly upregulated relative to those of the parental cells, implying that they play a significant role in UCB with acquired chemoresistance.

### 3.2. Targeted Inhibition of OGT with RNA Interference-Mediated Gene Silencing Restores Chemosensitivity and Attenuates the Oncogenic Potential of Chemoresistant UCB Cell Models

To verify the contribution of OGT to chemoresistance, UMUC-3-GEMr and T24-PTXr cells were transfected with shOGT to conduct loss-of-function experiments. *OGT* silencing (Figure 3A) resulted in a marked increase in chemosensitivity, reducing the IC_50_ of GEM from 123 μm to 54 μm in UMUC-3-GEMr cells and that of PTX from 4.4 μm to 2.6 μm in T24-PTXr cells (Figure 3B). In particular, the combination of the corresponding chemo-drugs (GEM, 7 μm; and PTX, 0.2 μm) with OGT-KD (Figure 3C) resulted in greater severe cell growth defects in the chemoresistant cells (UMUC-3-GEMr and T24-PTXr) compared with those observed following single treatment (Figure 3D).

Additionally, downregulation of O-GlcNAc induced by *OGT* silencing drastically suppressed cell migration (Figure 4A) (*p* < 0.001) and invasion (Figure 4B) (*p* < 0.05) in the UMUC-3-GEMr and T24-PTX4 sublines. Moreover, *OGT* silencing retarded the in vivo tumor growth capability of UMUC-3-GEMr cells (Figure 4C) through the suppression of cell proliferation (Ki-67) and induction of apoptosis (caspase-3), even without the addition of GEM (Figure 4D), which further suggests the additional tumor-promoting effects of OGT in chemoresistant UCB.

### 3.3. Comparative Transcriptome Analysis Provides Insights into Mechanisms Underlying OGT-Mediated Tumor Aggression and Chemoresistance in UCB

To elucidate the mechanisms underlying complex biological/molecular changes associated with acquired chemoresistance in UCB, we used RNA sequencing (RNA-seq) to assess differential expression across the transcriptome of chemoresistant UCB cells (UMUC-3-GEMr and T24-PTXr) and their parental cells (UMUC-3 and T24). The volcano plot showed 1243 (568 upregulated and 675 downregulated) and 1189 (478 upregulated and 711 downregulated) significant differentially expressed transcripts in UMUC-3-GEMr and T24-PTXr cells, respectively (Figure 5A). To explore common gene expression phenomena among drug-resistant cell lines, we focused on 97 genes that were consistently upregulated in both cell lines relative to the parental cells (Figure 5B), all of which are listed in Appendix A. As shown in Figure 5B and Appendix A, these 97 genes were significantly enriched in eight KEGG pathway terms, including the “Rap1 (Ras-proximate-1) signaling pathway” (7 genes), “Pathways in cancer” (9 genes), “Ras signaling pathway” (7 genes), “phosphatidylinositol 3-kinase (PI3K)-protein kinase B (AKT) signaling” (8 genes), “Focal adhesion” (6 genes), “Mitogen-activated protein kinase (MAPK) signaling pathway” (6 genes), “Extracellular matrix (ECM)-receptor interaction” (4 genes), and “Protein digestion and absorption” (4 genes) (*p* < 0.05 and FDR < 0.2).

OGT is the only enzyme that modulates O-GlcNAcylation in human cells. Therefore, to narrow down 97 genes to O-GlcNAcylation substrates associated with chemoresistance in UCB, we then predicted the OGT target candidates harboring potential O-GlcNAcylation sites by calculating the O-GlcNAcylation prediction score of the 97 genes (Appendix A). We finally selected four intersected genes (*VCAN*, *S1PR3*, *PDGFRB*, and *PRKCG*) as the primary genes involved in the enriched KEGG pathways with high O-GlcNAcylation prediction scores based on their potential relationship to OGT-mediated MDR (Appendix A). Notably, we observed significant increases in the mRNA expression levels of *VCAN*, *S1PR3*, *PDGFRB*, and *PRKCG* in UMUC-3-GEMr cells relative to levels observed in parental UMUC-3 cells, with this observation consistent with the RNA-seq data (Figure 5C) (*p* < 0.05). Immunoprecipitation analysis of O-GlcNAcylation of endogenous PDGFRB and PRKCG in UMUC-3-GEMr cells also supported that both PDGFRB and PRKCG might be important OGT substrates conferring acquired chemoresistance in UCB (Figure 5D). Furthermore, downregulation of VCAN, S1PR3, PDGFRB, and PRKCG levels via two independent sets of siRNA silencing in UMUC3-GEMr cells induced cell growth defects (Figure 5E), which could be downstream factors that contribute to OGT-mediated acquired chemoresistance in UCB.

## 4. Discussion

Several types of cytotoxic systemic chemotherapies are commonly used for either induction or salvage therapy with the goal of providing long-term complete responses in locally advanced or metastatic UCB [1,2,3,4,5,6,7,8,9,10,11,12,13,45]. GEM with cisplatin or methotrexate, vinblastine, adriamycin, and cisplatin (MVAC) has been the first-line chemotherapy regimen; unfortunately, these platinum-based combinations fail to provide sustained responses and are only applicable in a small proportion of patients [1,2,3,4,5,6,7,8,9,10,11,12,13,45]. Although alternative single or multiagent chemotherapy regimens, including GEM, PTX, pemetrexed, and vinflunine, are often used as second-line strategies, these regimens provide even lower efficacy, leading to significant morbidity and potential mortality [45]. Recently, a new class of immunotherapeutic agents called immune checkpoint inhibitors (ICI), such as atezolizumab, avelumab, and pembrolizumab, have shown potential in eliciting durable and complete responses in patients with locally advanced or metastatic UCB, in those progressing during or after platinum-based chemotherapy, or unfit for platinum-based chemotherapy [45]. However, the clinical efficacy of ICI is low, eliciting responses only in a small proportion of patients [45].

A growing number of studies suggest aberrant O-GlcNAc cycling as an important regulator of cancer progression and metastasis [24,25,26,27,28,30]. A previous study demonstrated that hyper-O-GlcNAcylation in UCB enhances the DNA damage response in addition to oncogenic phenotypes, and reducing hyper-O-GlcNAcylation by OGT-KD facilitates the chemosensitivity of UCB cells to the DNA-damaging agent cisplatin [27]; however, there is limited information describing the pathological role of OGT in the development of MDR in UCB. OGT, therefore, presents an attractive subject for study to potentially expand the treatment options for patients with re-emerging tumors after treatment with GEM and PTX for UCB. We employed a model for studying resistance to GEM or PTX in UCB using two initial cell lines (UMUC-3 and T24) to provide novel insights into the molecular mechanisms involved in enhanced chemoresistance and tumor aggression in UCB by elevated expression of OGT and O-GlcNAc. We found that the acquired resistance to GEM or PTX induced the overexpression of OGT in human UCB cell lines, and loss-of-function experiments verified that eliminating OGT function restored chemosensitivity to GEM or PTX, which suggests that O-GlcNAcylation by OGT in UCB contributes to the MDR and cancer aggression through direct or indirect impacts on various oncogenic mechanisms, although the detailed targeted proteins and mode of action remain to be further characterized (Figure 6). These preclinical findings suggest that the development of a novel treatment option using either cancer gene therapy with anti-OGT siRNA alone or in combination with GEM or PTX for UCB patients may be efficacious.

Cisplatin-based neoadjuvant chemotherapy (NAC) and radical cystectomy (RC) for muscle-invasive UCB are widely recognized as the optimal approach associated with improved overall and cancer-specific survival [46,47]. In a recent study, a nomogram predicting the 5-year UCB-specific mortality for patients with muscle invasive UCB undergoing cisplatin-based NAC and RC was constructed. The pathological parameters from the post-NAC RC specimens comprised the pathological T stage, ypN stage, and surgical margin status, which are key factors for determining survival probability and stratifying risk [48]. This is consistent with previous findings that the degree of downstaging after NAC is associated with the response to cisplatin-based NAC and is a critical determinant of subsequent survival [49,50]. Although this nomogram could be a useful tool for selecting at-risk patients for treatment intensification and adjuvant therapy [48], it does not predict whether the patients will respond to any specific available treatment options. Therefore, the optimal selection of patients for NAC, immediate RC, or other future treatment options will require further investigation of predictive biomarkers. Interestingly, Rozanski et al. [51] detected *OGT* mRNA in the urine of patients with UCB, but not in that of healthy patients, and OGT levels were higher in high-grade UCB than lower-grade tumors [51]. In future studies, it will be interesting to determine whether the content of OGT and O-GlcNAc in tumor tissues, urine, or blood could represent useful predictive biomarkers of disease progression and clinical outcomes in advanced UCB patients treated with chemotherapeutic agents.

Furthermore, the fundamental mechanisms underlying OGT-mediated chemoresistance in UCB should be also elucidated and validated through comprehensive gene expression profiling and quantitative glycoproteomic analysis for the global identification of OGT interactors that play important roles in promoting chemoresistance and tumor aggression in UCB. A larger panel of drug-resistant UCB cell lines could be further expanded to elucidate the underlying mechanism of OGT-mediated chemoresistance in UCB in conjunction with new patient-derived xenografts that show greater intratumoral heterogeneity and recapitulate clinically relevant responses to chemotherapeutics. Unfortunately, pharmacological silencing was not conducted in this study due to the current absence of commercially potent and selective OGT inhibitors. Although the OSMI compound series represents one of the most potent classes of OGT inhibitors reported to date, they were not found to be effective in suppressing OGT activity in our preclinical models compared with RNA interference-mediated OGT targeting (data not shown).

Given the key roles played by OGT substrates as well as their downstream effectors, it is unsurprising that increased levels of OGT/O-GlcNAcylation directly or indirectly impact the process of cancer metastasis by promoting anchorage-independent growth, migration, and invasion abilities of cancer cells [24,28]. There are several potential mechanisms by which OGT-mediated O-GlcNAcylation of substrate proteins modulate cancer progression, and they include the following cellular processes: (1) creating recognition sites for recruitment to initiate cascades, leading to the activation of downstream effectors, (2) cross-talk with PTMs to modulate substrate stabilization and activation, and (3) integration of cancer cell proliferation/metastasis/epithelial–mesenchymal transition (EMT)/stemness/metabolic alteration/angiogenesis-associated protein activities [24,25,26,27,28,29,30,31,52,53,54,55,56] (Figure 6). Notably, aberrantly activated RAS/MAPK and PI3K/AKT pathways reportedly promote chemoresistance and metastatic progression in advanced UCB [18,23,33,57,58,59], and this oncogenic pathway activation also mediates an increase in OGT and overall O-GlcNAcylation in several cancers through increased glycolysis [24,57]. For example, nuclear translocation and protein stability of YES-associated protein 1 (YAP1) is regulated by O-GlcNAcylation, and elevated YAP1 expression is associated with antiapoptotic effects, migration/invasion, EMT, and cancer stemness in UCB [25,54,55].

Commonly enriched gene sets in established chemoresistant UCB cell models support those of a series of studies indicating that the migratory/invasive potential of tumors is a key regulator of MDR and the basis of tumor cell metastasis in various cancer types, including UCB [23,60,61,62,63]. Interestingly, the most enriched gene sets in the UMUC3-GEMr and T24-PTXr sublines were related to the Rap1 and RAS signaling pathways. Rap1, a significant mediator of RAS functions, is now recognized as a central regulator of cancer cell invasion and metastasis via the activation of extracellular signal-regulated kinase (ERK), AKT, focal adhesion kinase (FAK), and Wnt signaling [64]. Forced increase in glucose uptake and metabolism activate multiple oncogenic pathways through Rap1, leading to the acquisition of a cancer phenotype in nonmalignant breast cells [65]. Moreover, a recent PPI analysis found that OGT-interacting proteins are related to anchoring and adherence junctions, which implies that OGT likely participates in cell adhesion and invasion [29]. In support of our findings, ECM–receptor interaction and focal adhesion are reportedly related to the acquisition of resistance to GEM or cisplatin in addition to the adhesive and invasive behavior of UCB cells [23,34,61,66,67]. Focal adhesion complexes facilitate cell–ECM contact, and the connection between ECM and the actin cytoskeleton essentially contributes to tumor cell resistance to different treatment regimens through focal adhesion-signaling hubs comprising integrins, growth factor receptors, and intracellular molecules [68,69]. Furthermore, EMT has been linked to metabolic reprogramming [70], which is supported by previous findings that mesenchymal cells exhibit a high rate of glycolysis that fuels cytoskeletal remodeling [64,71].

These findings provide chemoresistant cell models for use in acquiring a better understanding of the mechanisms underlying chemoresistance and for exploring potential biomarkers for GEM and PTX response in UCB patients. We identified four candidates, including VCAN, S1PR3, PDGFRB, and PRKCG, based on their highest O-GlcNAcylation-prediction scores following RNA-seq analysis. Unfortunately, we could not directly investigate the chemosensitizing effects of RNA interference-mediated silencing of VCAN, S1PR3, and PDGFRB, and PRKCG because KD of these genes alone induced a dramatic growth inhibition in UMUC-3-GEMr cells, supporting their respective roles in OGT-mediated tumor aggressiveness in chemoresistant UCB. VCAN is a member of the large chondroitin sulfate proteoglycan family with hyaluronate-binding capacities located in the ECM and inhibits apoptosis and exerts functions in metastasis through the ECM remodeling pathway in multiple types of cancers [72,73,74]. A previous study showed that the cisplatin-induced upregulation of VCAN contributes to aggressive biological behavior and poorer outcomes in patients with advanced BC [73]. Additionally, the S1P/S1PR3 axis plays a role in promoting cancer cell proliferation and aerobic glycolysis as well as inhibition of apoptosis by activating the YAP/c-MYC/phosphoglycerate mutase 1 axis in osteosarcoma cells [75]. Furthermore, S1P/S1PR3/Notch signaling activation results in the expansion of cancer stem cells (CSC) in several types of cancers [76]. Finally, PRKCG encodes protein kinase C gamma (PKCγ), which plays an important role in tumorigenesis, proliferation, differentiation, and migration [77,78,79,80]. For example, PKCγ interacts with fascin and Rac at the edge of cells to promote the migration of colon cancer cells [81], and the syndecan-2 cytoplasmic domain regulates matrix metalloproteinase-7 expression through PKC-mediated activation of FAK/ERK signaling [77]. Deregulation of PDGF-BB/PDGFR signaling is implicated in driving EMT, CSC self-renewal, metastatic potential, and chemoresistance in several cancers, including UCB, via activation of the PI3K and MAPK signaling pathways [82,83,84,85,86,87,88]. Further in-depth studies are needed to sufficiently validate the roles of VCAN, S1PR3, PDGFRB, and PRKCG in OGT-mediated chemoresistance in UCB and determine the detailed molecular mechanisms by which these four molecules regulate acquired chemoresistance in UCB.

In summary, these results show that hyper-O-GlcNAcylation and the deregulated expression of OGT could serve as novel potential therapeutic candidates for overcoming acquired chemoresistance to GEM and PTX in UCB.

## Figures and Tables

**Figure 1 biomedicines-10-01162-f001:**
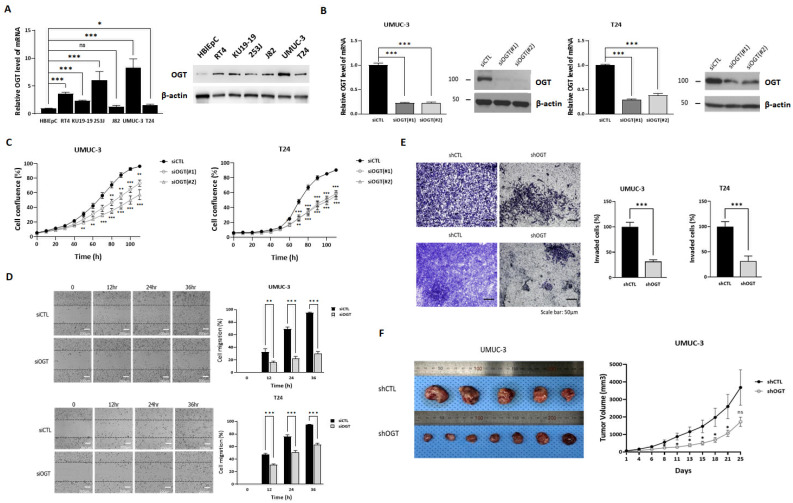
Downregulation of O-linked β-N-acetylglucosamine (O-GlcNAc) transferase (OGT) in urothelial carcinoma of the bladder (UCB) cells via gene targeting attenuated cell growth, cell migration and invasion, and tumor growth in vivo. (**A**) Western blot (WB) and qRT-PCR examination of OGT expression in human bladder epithelial cells and UCB cells. (**B**) WB and qRT-PCR examination of OGT expression in UMUC-3 and T24 UCB cells reverse transfected with control siRNA (siCTL) or siOGT. (**C**) UMUC-3 and T24 cells were reverse transfected with siCTL or two pairs of siOGT and incubated in the IncuCyte^TM^ analyzer to monitor cell proliferation. Cell confluence levels were measured in real time using the IncuCyte^TM^ analyzer and are presented as a percentage. (**D**) Wound healing assay of cell migration in UMUC-3 and T24 cells reverse transfected with siCTL or siOGT. Migrated cell images were acquired with Operetta CLS at 0, 12, 24, and 36 h. (**E**) Boyden Chamber assay analysis of the in vitro invasion capability of UMUC-3 and T24 cells stably expressing shCTL or shOGT. Cells were incubated for 48 h. (**F**) Tumor volume over time in nude mice injected subcutaneously with 5 × 10^6^ shCTL or shOGT UMUC-3 cells. Tumors were measured when average tumor size reached 40 mm^3^. After 4 weeks, the tumors were removed for morphological and histological examination. Representative images of tumors from shCTL (top) and shOGT cells (bottom) are shown. In panels (**A**–**E**), the data represent at least three independent experiments and error bars indicate mean ± SEM. * *p* < 0.05; ** *p* < 0.01; *** *p* < 0.001.

**Figure 2 biomedicines-10-01162-f002:**
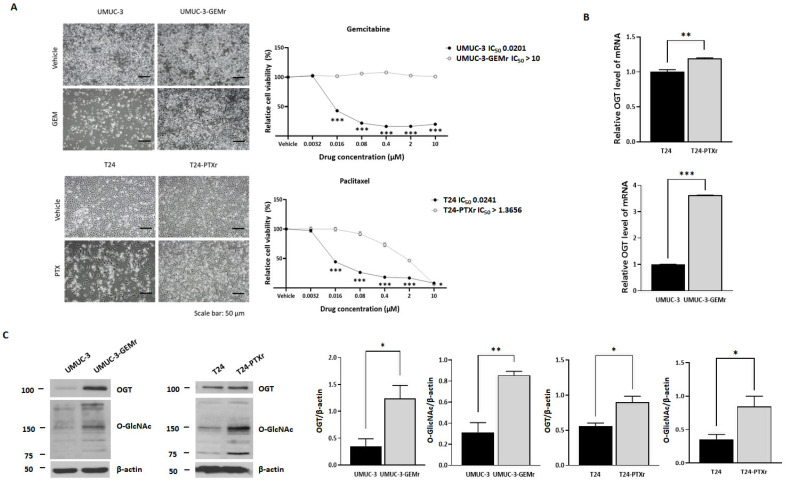
OGT expression and activity was upregulated in UCB cell models with acquired chemoresistance to gemcitabine (GEM) and paclitaxel (PTX). (**A**) Cell morphology of resistant and nonresistant cells observed under a light microscope (magnification, ×10). UMUC-3 and UMUC-3-GEM-resistant (GEMr) cells were treated with a vehicle or 10 μM GEM. T24 and T24-PTX-resistant (PTXr) cells were treated with a vehicle or 1 μM PTX. Observations were made 72 h after treatment. After observation, cell viability was analyzed by MTT assay. The data represent at least three independent experiments and error bars indicate the mean ± SEM. (**B**) qRT-PCR was used to detect OGT and GAPDH expression in UMUC-3/UMUC-3-GEMr and T24/T24-PTXr cells. Error bars indicate the mean ± SEM for three independent regions each. (**C**) Immunoblots of lysates derived from UMUC-3, UMUC-3-GEMr, T24, and T24-PTXr cells against OGT and O-GlcNAc. β-actin was used as the loading control. Error bars indicate the mean ± SEM for three independent regions each. For all panels * *p* < 0.05; ** *p* < 0.01; *** *p* < 0.001.

**Figure 3 biomedicines-10-01162-f003:**
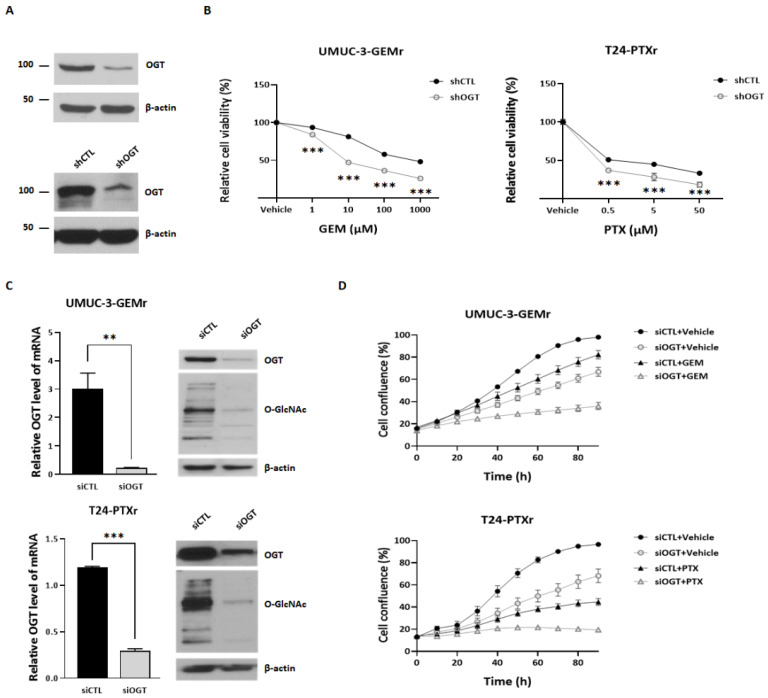
Knockdown of OGT restored sensitivity to chemotherapy drugs in UCB cells with acquired chemoresistance. (**A**) WB examination of OGT expression in UMUC-3-GEMr and T24-PTXr cells stably expressing shCTL or shOGT after incubation for 48 h. The data represent at least three independent experiments. (**B**) Cell viability analysis by MTT assay. UMUC-3-GEMr and T24-PTXr cells stably expressing shCTL or shOGT were treated with a vehicle or the corresponding drugs for 72 h. The data represent at least three independent experiments and error bars indicate the mean ± SEM. *** *p* < 0.001. (**C**) Immunoblot and qRT-PCR detection of OGT knockdown in UMUC-3-GEMr and T24-PTXr cells reverse transfected with siCTL or siOGT. UMUC-3-GEMr and T24-PTXr cell lysates were immunoblotted against OGT and O-GlcNAc. β-actin was used as the loading control. qRT-PCR was used to detect OGT and GAPDH expression in UMUC-3-GEMr and T24-PTXr cells. Error bars indicate the mean ± SEM for three independent regions each. ** *p*  <  0.01; *** *p* < 0.001. (**D**) UMUC-3-GEMr and T24-PTXr cells were reverse transfected with siCTL or siOGT. After 24 h, UMUC-3-GEMr cells were treated with a vehicle or 7 μM GEM and T24-PTXr cells were treated with a vehicle or 200 nM PTX. Cells were incubated in the IncuCyte^TM^ analyzer to monitor cell proliferation. Cell confluence levels were measured by the IncuCyte^TM^ analyzer in real time and are presented as a percentage. The data represent at least three independent experiments and error bars indicate the mean ± SEM.

**Figure 4 biomedicines-10-01162-f004:**
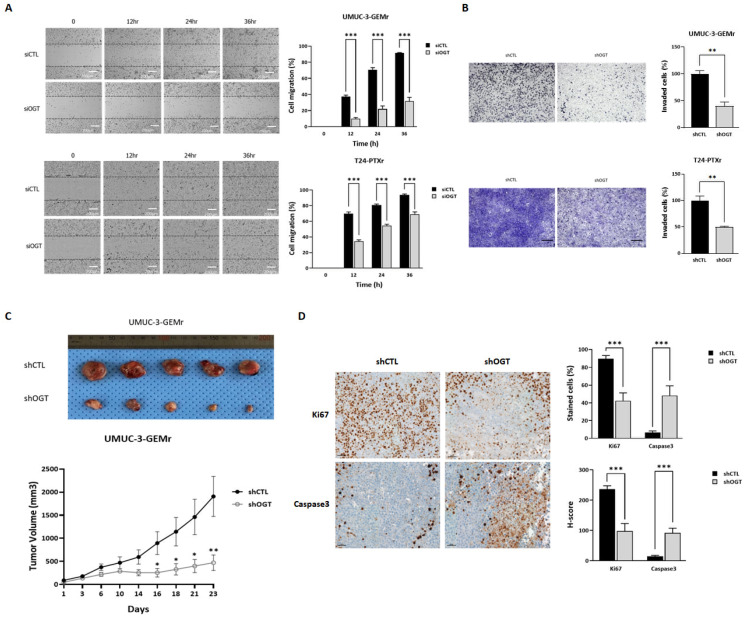
Downregulation of OGT via gene targeting in chemoresistant UCB cells suppressed cell migration, invasion, and subcutaneous xenograft tumor growth via the inhibition of cell proliferation and the induction of cell apoptosis. (**A**) UMUC-3-GEMr and T24-PTXr cells were reverse transfected with either siCTL or siOGT and cell migration was examined using a wound healing assay. Images of the migrated cells were acquired with Operetta CLS at 0, 12, 24, and 36 h. The data represent at least three independent experiments and error bars indicate the mean ± SEM. Scale bar, 200 μm. (**B**) The in vitro invasion capability of UMUC-3-GEMr and T24-PTXr cells was analyzed using the Boyden Chamber assay. UMUC-3-GEMr and T24-PTXr cells stably expressing shCTL or shOGT were incubated for 48 h. The data represent at least three independent experiments and error bars indicate the mean ± SEM. Scale bar, 50 μm. (**C**) Representative images of tumors from shCTL (top) and shOGT cells (bottom). The mice were injected subcutaneously with UMUC-3-GEMr cells stably expressing shCTL or shOGT (5 × 10^6^/mouse). Tumor size was measured at the indicated time points and, after 4 weeks, the tumors were removed for morphological and histological examination. (**D**) Representative images of immunohistochemistry from tumor tissues and quantified H-scores of Ki-67 and Caspase 3 from tumor tissues. Tumors derived from shCTL or shOGT UMUC3-GEMr cells were dissected 6 weeks after cell inoculation and embedded in paraffin, and tumor sections were stained with antibodies against Ki-67 or Caspase 3. Error bars indicate the mean ± SEM for three independent regions each. Scale bars, 100 μm. For all panels * *p* < 0.05; ** *p*  <  0.01; *** *p* < 0.001.

**Figure 5 biomedicines-10-01162-f005:**
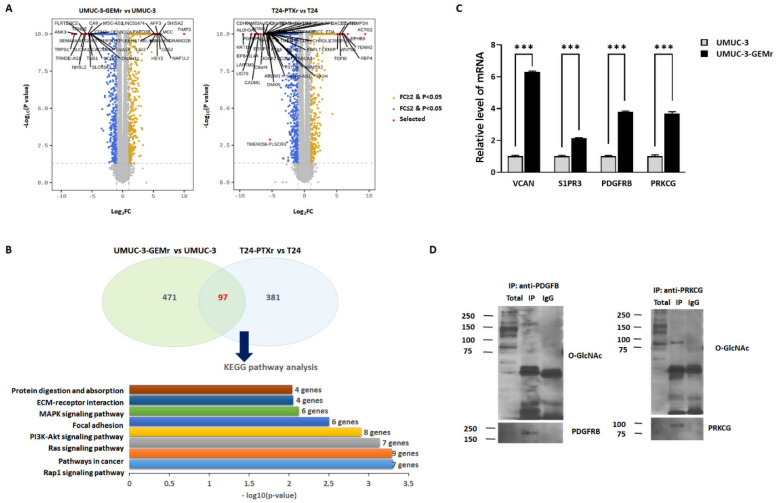
Preliminary investigation of the mechanisms of chemoresistance in UCB cells using comparative gene expression analysis. (**A**) Volcano plot of significantly differentially expressed (SDE) genes in UMUC-3-GEMr and T24-PTXr cells compared to their parental cells. Yellow and blue dots indicate significantly up- and downregulated genes in chemoresistant cells, respectively. In UMUC-3-GEMr cells, 568 genes were upregulated, and 675 genes were downregulated; in T24-PTXr cells, 478 genes were upregulated, and 711 genes were downregulated. FC: fold change. (**B**) Common upregulated gene sets shared by two sets of UMUC-3-GEMr and T24-PTXr cells. Kyoto Encyclopedia of Genes and Genomes (KEGG) pathway enrichment analysis was run based on consistent SDE genes from the upregulated genes in UMUC-3-GEMr or T24-PTXr cells. (**C**) RT-qPCR detected VCAN, S1PR3, PDGFRB, and PRKCG expression in UMUC-3 and UMUC-3-GEMr cells. Error bars indicate the mean ± SEM for three independent regions each. *** *p* < 0.001. (**D**) Detection of O-GlcNAc modifications of endogenous PDGFRB and PRKCG in UMUC-3-GEMr cells. Cells were treated with 2 μM Thiamet G for 6 h before lysis. Lysates were immunoprecipitated with either anti-PDGFRB antibody or anti-PRKCG antibody, then immunoblotted against PDGFRB or PRKCG, and O-GlcNAc. (**E**) UMUC-3-GEMr cells were reverse transfected with either siCTL or two independent sets of siVCAN, S1PR3, PDGFRB, and PRKCG and incubated in the IncuCyte^TM^ analyzer to monitor cell proliferation. Cell confluence levels were measured using the IncuCyteTM analyzer in real time and are presented as a percentage. The data represent at least three independent experiments and error bars indicate the mean ± SEM. * *p*  <  0.05; ** *p*  <  0.01; *** *p * <  0.001.

**Figure 6 biomedicines-10-01162-f006:**
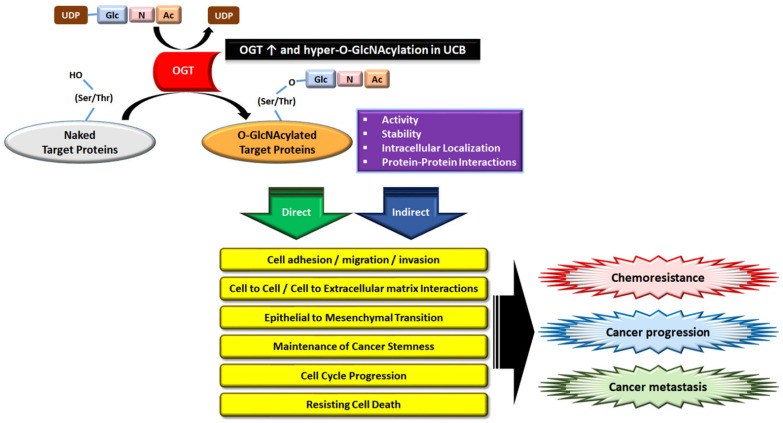
Graphical representation of potential models of enhanced chemoresistance and tumor aggressiveness mediated by upregulated OGT activity and hyper-O-GlcNAcylation in UCB. Glc, glucose; UDP-GlcNAc, uridine diphosphate N-acetylglucosamine. O-GlcNAcylation is a dynamic post-translational modification occurring on serine (Ser) and threonine (Thr) residues of nuclear and cytoplasmic target proteins.

## Data Availability

The datasets generated and/or analyzed during the current study are available from the corresponding authors on reasonable request. The sequence data in this publication were deposited in the NCBI Sequence Read Archive under the accession number PRJNA805282.

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
