# Peer review of "Targeted Inhibition of O-Linked β-N-Acetylglucosamine Transferase as a Promising Therapeutic Strategy to Restore Chemosensitivity and Attenuate Aggressive Tumor Traits in Chemoresistant Urothelial Carcinoma of the Bladder"

_biomedicines, 2022, doi:10.3390/biomedicines10051162_

Round 1

Reviewer 1 Report

The author's response still can't answer my concern. 
Although higher OGT level correlated with chemoresistance, you can't explain the OGT-mediated chemoresistance 
by comparing the chemoresistant cells with their parental cells. More contributive mechanisms might exist for their chemoresistance.
If no further proof/experiments were done, either the title of figure 5 should revised or this data should not be included in this manuscript.

Author Response

The author's response still can't answer my concern. Although higher OGT level correlated with chemoresistance, you can't explain the OGT-mediated chemoresistance by comparing the chemoresistant cells with their parental cells. More contributive mechanisms might exist for their chemoresistance. If no further proof/experiments were done, either the title of figure 5 should revised or this data should not be included in this manuscript.

Response: We completely agree with the reviewer’s comment and thank for valuable suggestion. Per suggestion, the title of Figure 5 was modified to “Preliminary investigation of the mechanisms of chemoresistance in UCB cells using comparative gene expression analysis.” (Page 13 of 23, line 475-476). The changed part has been highlighted in yellow.

Reviewer 2 Report

Authors present an interesting analysis exploring an important field of urological oncology. I have few comments for this overall well written manuscript:

  1. Did authors adjusted their analyses for multiple hypothesis testing? Did they rely on false discovery rate to sdjust analyses? The statistical analyses within methods section should be discussed deeper
  2. I would add a figure illustrating the network involving the examined pathways and their role in urothelial cancer pathofisiology
  3. Authors might discuss their results with a clinical prespective that include the possibility to use also other treatments than platinum-bbased. I would suggest to include in the discussion findings resumed by Marchioni et al. in their review of the literature (10.1080/14737140.2018.1439744.)
  4. Classical predictors of response to platinum-based regimen have been tested. Authors should discuss such predictors on the light of their findings (10.1016/j.euf.2020.07.002.)

Reviewer 3 Report

The work systematically studied the role of O-linked β-N-acetylglucosamine transferase (OGT) in acquired resistance of urothelial carcinoma of the bladder (UCB) to gemcitabine and paclitaxel and how its downregulation restored the sensitivity to chemotherapy drugs in UCB cells with acquired chemoresistance. The work reported in this manuscript is interesting and well presented. However, it requires corrections and improvements before acceptance. The work requires revision. Some comments are:

  1. Provide Quantitative data like the IC50 values in the ‘abstract’ on how OGT downregulation restored the chemosensitivity to improve the representation of key results in the abstract.
  2. Have the Authors examined for any morphology changes in established chemoresistance UCB cell sublines compared to the original cell line used to generate?
  3. There are many grammatical and sentence errors in the article, and the language organization needs to be improved

Round 2

Reviewer 1 Report

No further comment